

# The High Intensity Proton Accelerator Facility

**J. Grillenberger⋆, C. Baumgarten and M. Seidel**

Paul Scherrer Institut, 5232 Villigen PSI, Switzerland

⋆ joachim.grillenberger@psi.ch

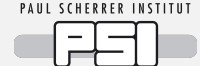

## Abstract

The High Intensity Proton Accelerator Facility at PSI routinely produces a proton beam with up to $1.4\,\mathrm{MW}$ power at a kinetic energy of $590\,\mathrm{MeV}$. The beam is used to generate neutrons in spallation targets, and pions in meson production targets. The pions decay into muons and neutrinos. Pions and muons are used for condensed matter and particle physics research at the intensity frontier. This section presents the main physics and technology concepts utilized in the facility. It includes beam dynamics and the control of beam losses and activation, power conversion, efficiency aspects, and performance figures, including the availability of the facility.

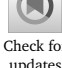

## 2.1 Introduction

The original proposal for the accelerator facility that is now known as the PSI high intensity proton accelerator (HIPA)[1], was completed 1963 [2]. The objective was to produce a proton beam of several tens of microampere with an extraction rate higher than 50 % and an energy above 450 MeV, with the main goal to produce $\pi$-mesons and muons[2]. The final beam energy was later raised to $\geq$ 580 MeV and the specified beam current raised to $100\,\mu$A [3]. The main accelerator is the Ring cyclotron, an isochronous proton machine with eight separate magnet sectors and four main accelerating cavities operating at 50.6 MHz. The Ring cyclotron is designed to accelerate an injected 72 MeV proton beam to 590 MeV. The first pre-accelerator, called the Injector I cyclotron, was designed and constructed by Philips (Eindhoven). Injector I was a multi-purpose machine, that accelerated protons up to 72 MeV with a maximal extracted current of $I_{\max} \leq 180\,\mu$A, and also light ions for nuclear physics research. After one year of operation, in 1975, the highest beam current on target was $25\,\mu$A. The performance of the Ring cyclotron was steadily improved, especially the extraction efficiency. In December 1976 an extraction efficiency of 99.9 % (Ring) and of 85 % (Injector I) was achieved. The peak

---

[1]Formerly named the *Isochronous Cyclotron Meson Factory of ETH Zurich* [1], then the Schweizerische Institut für Nuklearforschung (S.I.N.) Ring Cyclotron.

[2]The term *meson* production targets was established for historical reasons - even though muons are leptons.

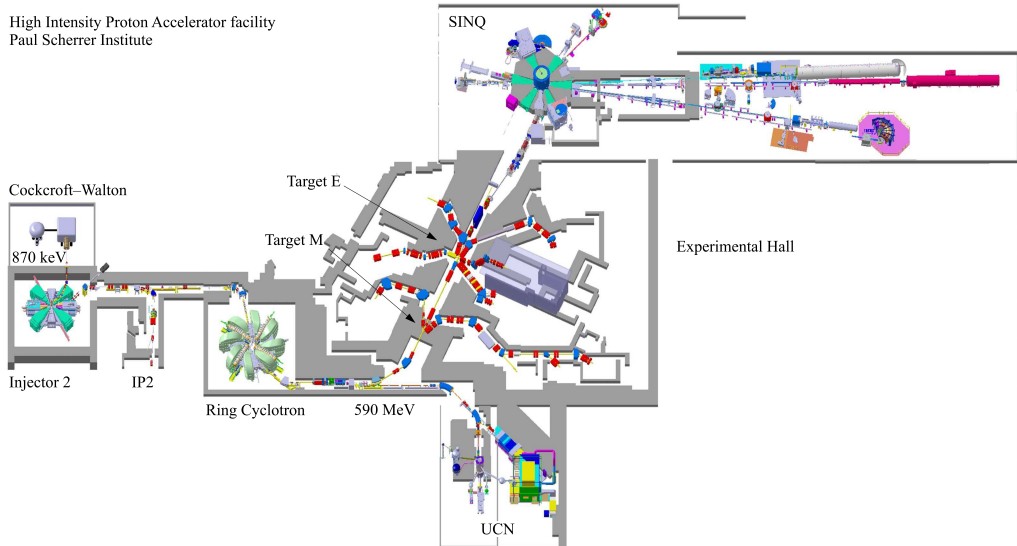

Figure 2.1: Layout of the High Intensity Proton Accelerator facility at the Paul Scherrer Institute.

intensity was raised within two years from $12\,\mu$A to $112\,\mu$A [4]. The beam current was limited by the 9 % beam losses at the extraction of Injector I and the resulting activation of components. Injector I was also used for low-energy experiments. During these experiments, Injector I was not available as a proton driver for the Ring cyclotron. Injector I was not able to deliver beam currents higher than about 180 $\mu$A (originally 100 $\mu$A specified), while the performance of the Ring cyclotron indicated the capability for much higher currents with low losses. Therefore, studies for an upgrade of the Ring cyclotron with a flattop cavity and a new injector cyclotron with a Cockcroft-Walton type pre-accelerator for beam currents of up to 1 mA were in progress while the commissioning was still ongoing [5]. At this stage, it was estimated that the Ring cyclotron had the potential to accelerate currents of up to $2-4$ mA [6]. The proposal to use two pre-accelerators, a 860 keV Cockcroft-Walton type accelerator followed by the new Injector II cyclotron, was approved in 1978.

Since 2010 the protons are produced by a compact small electron cyclotron resonance source with a 60 kV extraction system [7]. Two solenoids are used to focus the extracted protons onto a collimator. Hydrogen ions ($H_2^+$ and $H_3^+$), which are extracted as well, are only weakly focused due to their lower charge-to-mass ratio, and are stopped. The protons are accelerated in three stages. A Cockcroft-Walton DC linear accelerator, shown left in Figure 2.1, is used to pre-accelerate the DC proton beam to 0.87 MeV as required for the injection into the first turn of the Injector II cyclotron. The beamline is equipped with a bunching system a few meter upstream of the axial injection line, to match the beam phase space to the acceptance of Injector II. Injector II accelerates the pre-bunched beam with two high-voltage double-gap resonators[3] to an energy of 72 MeV within 80 turns. The extracted beam is then sent to an electrostatic beam splitter, where up to 100 $\mu$A can be split off for the production of radio-isotopes. The main part of the beam is injected into the Ring cyclotron with an electrostatic inflection channel. Eight normal-conducting magnets keep the particles' on their spiral path in the cyclotron. Four 50.6 MHz cavities accelerate the beam to its final kinetic energy of 590 MeV. After about 180 turns in the cyclotron, the beam is extracted with an electrostatic element (see Figure 2.2) and sent to the meson production targets [8]. These targets are

---

[3]A double-gap resonator is equivalent to a conventional Dee with two accelerating areas (gaps). In contrast the PSI Ring cyclotron uses hollow "single-gap" cavities.

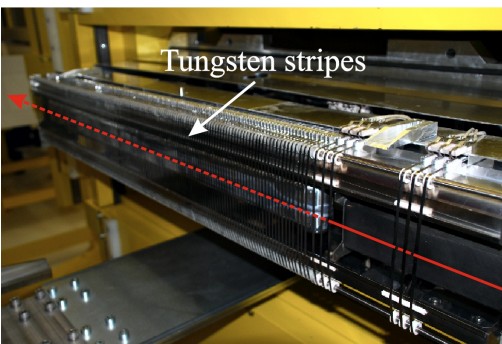 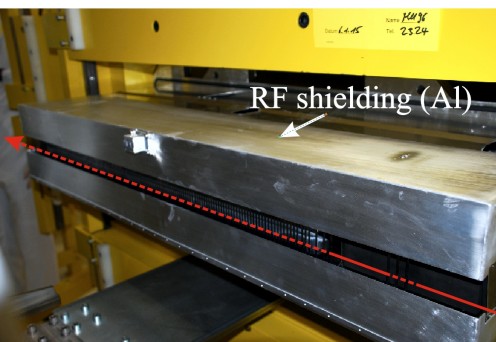

Figure 2.2: Pictures of the electrostatic extraction channel EEC without (left) and with attached aluminium shroud (right). The red arrow denotes the beam passing through the channel. The dashed part of the arrow denotes the parts where the beam passes through in between the grounded tungsten stripes and the aluminium cathode. The electric field of $8-10$ MV/m deflects the beam by 8 mrad on 920 mm effective length so it can be extracted from the cyclotron by a subsequent septum magnet.

made of graphite and limited in thickness so that the beam loses only a small fraction of its energy. After passing through a collimation system, needed due to multiple scattering in the meson production targets, roughly 60 (70) % of the beam current is left for a target thickness of 60(40) mm, and is then sent to the neutron spallation source SINQ [9–14]. If SINQ is not ready for beam, the beam is sent to the 590 MeV beam dump. Due to cooling issues, the beam current is then limited to 1.7(2.0) mA on a 40(60) mm thick meson production target. The Ultracold Neutron Source (UCN) is in operation since 2011 [15–19]. A fast kicker magnet just upstream of the meson production targets deflects the beam for a short time between 2 and 8 s to the UCN facility [20]. The duty cycle is restricted to a maximum of 3%.

The intention of this article is to present performance figures for the accelerator together with the main physics and technology concepts utilized in the facility. This includes beam dynamics and space charge effects in the cyclotrons, the control of beam losses and activation, power conversion, and efficiencies. While some of these topics are relevant only for cyclotrons, many themes are discussed that are important for any type of high intensity proton accelerator. In the following sections, the basic physics and main parameters of the three accelerators are described.

## 2.2 Injector II

The Injector II cyclotron was designed for high current operation, 1 mA and above, with minimal extraction losses. High extraction efficiency in a cyclotron demands a large turn separation. This can be achieved by the combination of high accelerating voltage, large radius, large gap magnets and low energy spread. To counter the strong defocusing space charge forces, a high vertical ("axial") betatron-tune[4] is required. Hence Injector II was designed as a low-field separate sector machine using four wedge sectors. The sector magnets leave space for two high-voltage double-gap resonators operating at the 10th harmonic of the orbital frequency and two single-gap flat-top resonators to minimize the energy spread. Since the injection energy of 870 keV is well below the Coulomb threshold, the first few turns can be used to collimate the beam and clean up halo [21].

---

[4]The "tune" is the number of vertical or horizontal oscillations of a particle per turn and characterizes the strength of vertical/horizontal focusing. Isochronous cyclotrons have, in contrast to synchrotrons, no intrinsic longitudinal focusing.

M.M. Gordon was the first to recognize that space charge in isochronous cyclotrons can lead to (as he called it) "vortex motion" [22]. Later Chabert, Luong and Promé as well as Chasman and Baltz backed this up theoretically [23, 24]. Numerical simulations, performed by Adam, Koscielniak, Adelmann and others, confirmed this effect [25–28]. The vortex effect can lead to increased halo formation and bunch "breakup". This has been experimentally investigated by Pozdeyev *et al* in the *small isochronous ring* (SIR) experiment [29]. The beam breaks up only if it is long initially and the breakup typically generates a number of self-sustaining round sub-bunches [29]. In case of a single initially short and compact bunch, the vortex effect stabilizes the bunch: the space charge induces a coupling between the longitudinal and horizontal motion that generates a weak (but non-zero) longitudinal focusing, an effect that can be understood with an analysis of the linear coupling terms of an isochronous cyclotron [30], although this is somewhat counter-intuitive. The usefulness of the self-focusing was discovered by the PSI operation crew, who achieved a high extracted current with low losses while the flat-tops were switched off by accident. Since the flat-top system was –with an appropriate setup– no longer required to achieve a low energy spread, the phase was reversed so as to operate in an accelerating mode. This enabled a further increase in the energy gain per turn and hence to reduce the turn number $N$. A maximum beam current of 2.7 mA has been extracted from Injector II on beam dump and 2.4 mA in combination with the Ring cyclotron.

The flat-top resonators will be replaced, in an ongoing upgrade program, by two 50 MHz high-voltage resonators. This should further reduce extraction losses and allow for even higher beam currents. However, the vortex effect generates bunches in a meta-stable state and is sensitive to various possible distortions [31, 32]. Making use of the vortex effect in Injector II may be possible due to the very conservative layout of the cyclotron, including a strict isochronism, [30] with a central region equipped with various movable collimators to optimize the bunch formation and to eliminate the halo [21]. Injector II is the only production cyclotron world-wide that is known to take advantage of the vortex effect.

## 2.3 The Ring Cyclotron

In 1975, after one year of operation, the highest beam current on target was 25 $\mu$A. The performance of the Ring cyclotron was steadily improved, especially the extraction efficiency. In the beginning, only a well-centered beam was able to pass the Walkinshaw-resonance without substantial beam loss, as the beam had to pass the resonance four times before extraction [5, 33]. A modification of the tune diagram by an improved setting of trim coils reduced this to two fast passages through the resonance and allowed relaxation of the requirement of beam centering [34, 35]. This enabled a considerable increase in the turn separation at extraction by means of precessionally-enhanced turn separation. In December 1976 an extraction efficiency of 99.9 % was achieved with a peak intensity of 112 $\mu$A [4]. Ten years later, after the first commissioning of the new pre-accelerators, a beam current of 1 mA was achieved with Injector II alone, and 310 $\mu$A in combination with the Ring cyclotron.

In 1981, Werner Joho presented an analysis of high intensity problems in cyclotrons [36], known as Joho's $N^3$-Law, which states that the loss dominated current limit $I_{\max}$ scales with the inverse third power of the number of turns $N$, $I_{\max} \propto N^{-3}$. This formula predicted the performance of the PSI Ring cyclotron of the following two decades with high accuracy [37, 38].

An upgrade of the RF system of the Ring was required and initiated for another substantial intensity increase [39]. In parallel, a bunching system was built and commissioned in the 870 keV injection line to better match the DC beam to the phase acceptance of Injector II [40, 41]. The upgrade of the RF system allowed a significant reduction of the number of turns in the Ring cyclotron and an increase of the production current to 2.2 mA (test-wise in dedicated shifts up to 2.4 mA) and the beam power to 1.3 MW (1.4 MW), in good agreement with Joho's

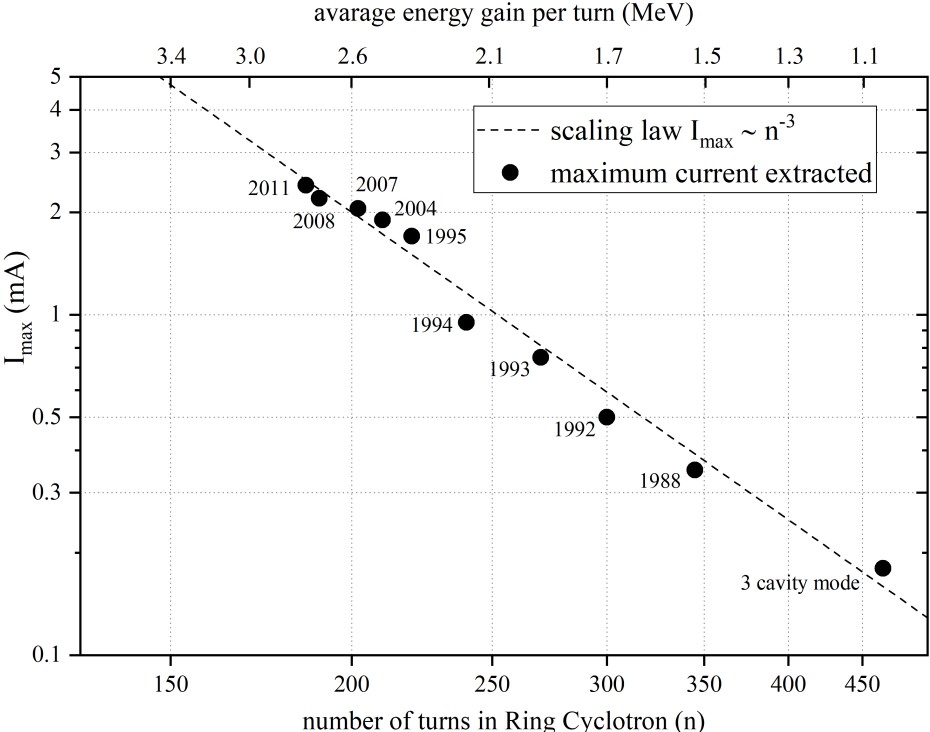

Figure 2.3: Joho's empirical law.

$N^3$-Law (see Figure 2.3). On full completion of the upgrade programs, which includes the replacement of the old 150 MHz flattop cavity, a beam current of 3 mA with a power of 1.8 MW should be within reach of both, Injector II [21] and the Ring cyclotron [42, 43].

## 2.4   Facility Performance

Every year, PSI has 1500-2000 user visits at the neutron source (SINQ), the muon source (S$\mu$S), and the facilities for particle physics (CHRISP) including the UCN Source. During more than 3000 instrument-days, over 800 experiments are performed each year. These user facilities are all part of the HIPA facility which operates at a beam power of up to 1.42 MW. In the following sections we describe the basic operation scheme of the facility and present the main details of the experimental stations. The performance of the accelerator, i.e., the achievable beam power, the availability, and its energy efficiency are also addressed.

### 2.4.1   Operation Scheme

A typical year of operation starts in the beginning of May after the shutdown and ends on Christmas with the next shutdown. The start of user operation may vary depending on the duration of the necessary maintenance and planned upgrade. The beam time schedule is compiled by the facility management in close collaboration with the user office of PSI. During regular user operation, the accelerators are operated nonstop for 24 hours the day. With the user operation starting in the beginning of May and ending at Christmas, the accelerator facility typically provides 200 days of primary beam for experiments. After three weeks of user operation, a maintenance period of two days is scheduled. In addition, two shifts of beam development before and after each maintenance are carried out to reduce beam losses and to improve the performance of the facility.

### 2.4.2 Pion and Muon Production

The production of pions and muons is possible with beam sent either to the spallation neutron target or to the beam dump. In the latter case, the maximum beam current extracted from the Ring cyclotron is limited to 1.7 mA due to the cooling limitations of the beam dump. Nevertheless, meson production is possible even though the spallation source may not be operational. The meson targets provide secondary particles for the experimental facilities. The performance of the meson facilities, i.e., the particle fluxes are given in Table 2.1.

Table 2.1: Particle types available at the meson experimental facilities. The rate is given in particles per second and per *mA* beam current and may vary with the selected momentum.

| Target (thickness) | User facility | Particle type | Momentum range (MeV/c) | max. rate ($s^{-1}mA^{-1}$) |
|---|---|---|---|---|
| M (5$mm$) | $\pi$M1 | e/$\pi$/$\mu$/p | $10 - 450$ | $2 \cdot 10^8$ |
| | $\pi$M3.1-3 | $\mu$ | $10 - 40$ | $3 \cdot 10^6$ |
| E (4 or 6$cm$) | $\pi$E1 | $\pi$/$\mu$/p | $10 - 450$ | $1 \cdot 10^9$ |
| | $\pi$E3 | $\mu$ | $10 - 40$ | $3 \cdot 10^7$ |
| | $\pi$E5 | $\pi$/$\mu$ | $10 - 120$ | $5 \cdot 10^8$ |
| | $\mu$E1 | $\mu$ | $60 - 120$ | $6 \cdot 10^7$ |
| | $\mu$E4 | $\mu$ | $10 - 40$ | $4 \cdot 10^8$ |

### 2.4.3 Neutron production

The main beam passes through the two graphite targets before striking the spallation neutron target of SINQ so it has to be collimated due to a five-fold increase in beam emittance. For an E-target thickness of 4(6) cm, about 70 %(60 %) of the beam current remains. The proton kinetic energy is degraded to 570 MeV (565 MeV). The remaining beam is first bent downwards and then sent back up vertically onto the spallation target. The thermal neutron flux scales with the beam current and is approximately $1.5 \cdot 10^{14}$ cm$^{-2}$s$^{-1}$ near the target.

The UCN facility was commissioned in 2010 and a measurement of the neutron electric dipole moment, nEDM, began in 2011. For this experiment, the full 590 MeV beam is switched periodically from the meson production targets to the UCN target with a fast-switching magnet. Typically, the beam is switched every 12 minutes for 8 seconds. Both the pulse duration and frequency can vary depending on the requirements of the experiments. This corresponds to a duty cycle of approximately 1 %. The pulse sequence is controlled by a software routine that decreases the beam intensity by 20 % roughly 2 s before the kick. After switching on the kicker magnet, the maximum intensity is then re-set to the nominal value during another 2 s. The reverse routine applies after the kick.

When the beam is switched back to the meson production and SINQ targets, the beam current is lowered below 1 mA and then raised back to the maximum within 20 s. This is done to avoid high thermal stress to the targets, particularly the SINQ-target.

### 2.4.4 Isotope Production

The Injector II cyclotron can produce 72 MeV protons for the production of radioactive isotopes. Two operating modes are possible: An electrostatic beam splitter can split off up to 100 $\mu$A of the main beam, which is directed to the isotope production target along a dedicated beamline. In this case, both the isotope production beam and main beam onto meson and

neutron production targets can operate simultaneously. Alternatively, the full beam, limited to $100\,\mu$A, can be sent to the isotope production target.

### 2.4.5 Accelerator Performance and Beam Intensity

The facility, originally designed for a maximum beam current of $100\,\mu$A, has continuously been improved to reach a maximum beam power of 1.42 MW, at present. The following section describes the performance characteristics of the accelerator facility, in particular the beam power and availability.

The maximum beam power is limited by the tolerable amount of proton losses during acceleration to meet legal obligations and to avoid activation and damaging of accelerator components. Currently, PSI is authorized to extract a maximum beam current of 2.4 mA from the Ring cyclotron, which has been achieved in the years 2011, 2012, 2015, and 2016. Furthermore, PSI may increase the beam current to a maximum of 2.6 mA during dedicated beam development shifts for eight hours every four weeks. Major steps in the increase of the beam power were achieved by replacing the Injector I cyclotron with the Cockcroft-Walton and Injector II pre-accelerators in 1985, and by continuous upgrades of the RF systems starting in 1990. Newly designed meson production targets have been used since 1991 to tolerate the thermal stress imposed by the higher beam power. After the commissioning of the spallation neutron target SINQ in 1996, the beam power was increased from 826 to 885 kW.

Following the installation of the fourth and last new copper cavity in the Ring cyclotron, the beam losses in the cyclotron were further reduced by increasing the peak voltage of each accelerating cavity from 790 MV to 850 MV. A maximum beam current of 2.4 mA was extracted on 20 June 2011 for the first time. The corresponding beam power of 1.42 MW was the highest ever achieved with any accelerator at that time. In Figure 2.4, the increase of the beam power for the years from 1974 to 2020 is shown.

The charge delivered on the meson and the neutron production targets scales with the average beam current extracted from the Ring cyclotron and is shown in Figure 2.5.

The beam intensity in HIPA is limited by beam losses. As practical experience has shown, the highest acceptable losses for hands-on maintenance are of the order of 100 W ($10^{-4}$ for 1 MW of beam power) per location. A major contribution is scattering of halo particles in the high voltage electrode of the extraction septum. Such losses are then distributed over several meters of beamline elements and lead to activation with maximum dose rates of the order of a few millisievers per hour. Such dose rates are acceptable for service work and handling components. For any further increase of the beam current, the relative losses in the cyclotron and the beam line would have to be reduced inversely proportional to the intensity to keep the activation at an acceptable level. The extremely high extraction efficiency of the PSI Ring cyclotron is a property that was optimized to allow the operation with high intensities. There are two key elements for low loss beam extraction: The generation of beam tails must be suppressed as best as possible, and the turn separation at the extraction septum must be maximized. In this way the density of halo particles at the position of the extraction septum is minimized. For an isochronous cyclotron the radial increment of the orbit radius per turn can be computed as

$$\frac{\mathrm{d}R}{\mathrm{d}n_\mathrm{t}} = \frac{U_\mathrm{t}}{m_0\,c^2}\frac{\gamma R}{(\gamma^2-1)\,\nu_r^2}. \tag{2.1}$$

$$= \frac{U_\mathrm{t}}{m_0 c^2}\frac{R}{(\gamma^2-1)\gamma}. \tag{2.2}$$

Here $\gamma$ is the relativistic energy factor, $\nu_r$ the radial tune, $U_t$ the energy gain per turn and $m_0$ the rest mass of the proton. Clearly a high acceleration voltage helps, but one finds a very strong

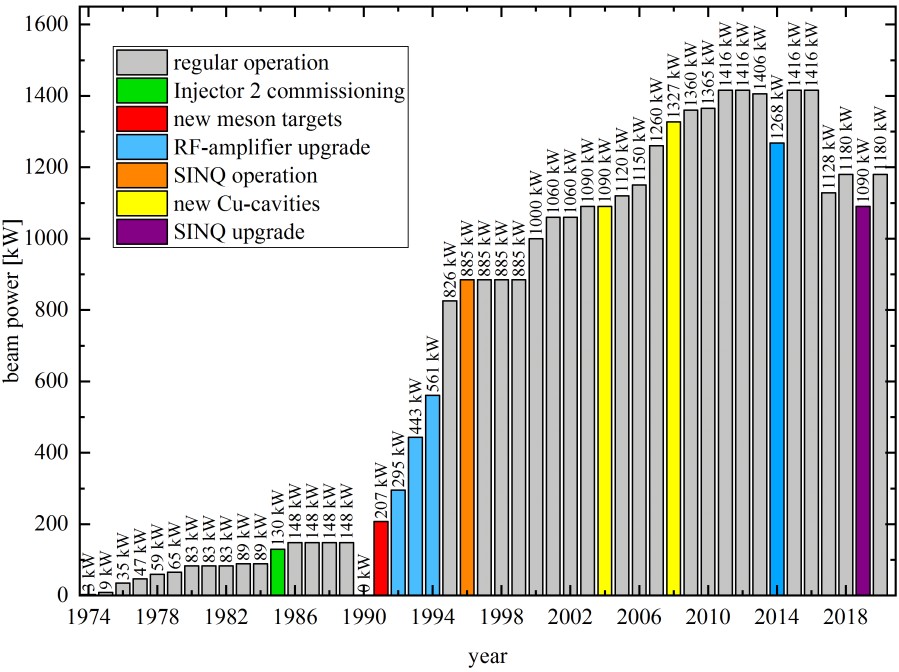

Figure 2.4: The maximum beam power achieved in the accelerator facility. In 1990 the facility was off line for the installation of new RF-amplifiers for the Ring cyclotron and the new meson production target station E including the beamline up to the beamdump.

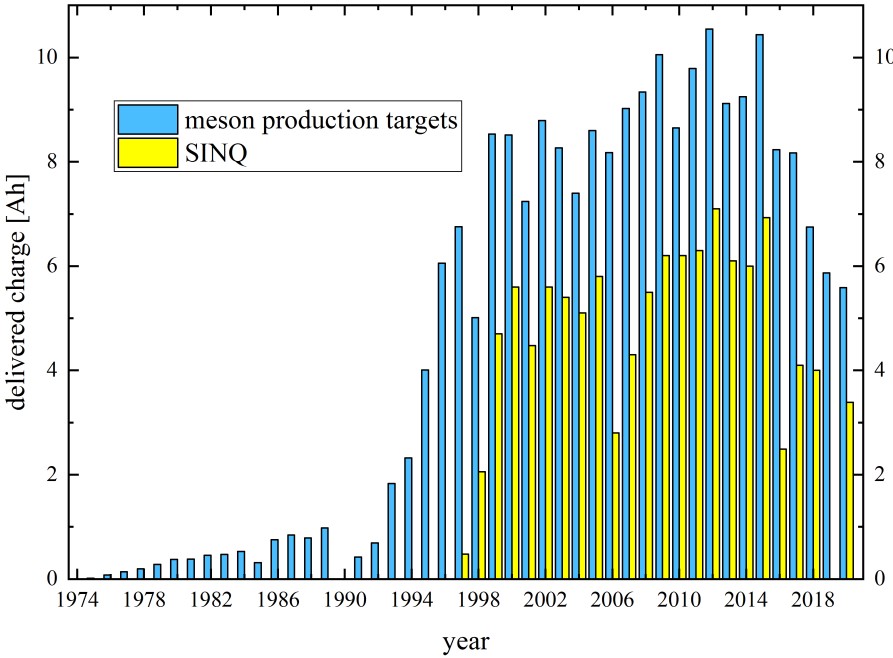

Figure 2.5: History of the charge delivered per year to the meson production targets and the neutron spallation target SINQ.

reduction with $\gamma$ for higher energies. Equation (2.1) illustrates the possibility to influence the turn separation by weaker focusing over the outer turns of the cyclotron. This violates the isochronous condition and is therefore only possible over a small number of turns. The second line (2.2) is the more general relation, for which $\nu_r \approx \gamma$. We also note the scaling with the extraction radius $R$, i.e. the size of the cyclotron. With an extraction radius of 4.5 m, the PSI Ring cyclotron is one of the largest cyclotrons in the world. An effective way to increase the turn separation at the extraction element is the introduction of orbit oscillations by deliberately injecting the beam slightly off centre. When the phase and amplitude of the orbit oscillation are chosen appropriately, and the behaviour of the radial tune is controlled in a suitable way, the beam separation can be increased by a factor of three. This gain is equivalent to a cyclotron three times larger and is thus significant. Figure 2.6 illustrates how this scheme is used in the PSI Ring cyclotron. In [44], the beam profile in the outer turns was computed numerically for realistic conditions, and the results are in good agreement with measurements.

In Figure 2.7 the frequency of beam losses at a certain current is depicted for the user operation at 2 mA in 2010 and at 2.2 mA in 2015.

## 2.5 Operating Statistics

High beam power is important for precise measurements of short duration. However, the availability of a large research facility is often of even greater importance to users. In this section, we describe beam time statistics and outage characteristics.

The availability of the HIPA facility requires a beam current of at least 1 mA extracted from the Ring cyclotron during scheduled user operation. According to this definition, the accelerator availability is 100 % if the beam current measured at the meson production target is equal or greater 1 mA. The lower limit of 1 mA is used to meet the needs of the experimental facilities, which require at least this current for performing meaningful measurements. A beam current of least 700 $\mu$A onto the spallation target is required for neutron experiments. This corresponds to 1 mA of beam current extracted from the Ring cyclotron. The lowest beam current considered as useful for the user community has been raised from 150 $\mu$A to 1000 $\mu$A in 2001. An outage of the spallation neutron target SINQ does not affect the availability of the accelerator since the collimated beam after the graphite targets can be sent onto the beam dump. Figure 2.8 shows the availability from 1974 to 2020.

A short interruption refers to outages lasting less than five minutes. The average number of short interruptions per year is roughly 15000, but it varies by more than a factor of seven as shown in Figure 2.9.

After the replacement of the first aluminium cavity with a copper cavity in the Ring cyclotron in 2005, major problems were experienced with the electrostatic elements in the cyclotron. Stable operation was not possible during the first month after the yearly shutdown. Frequent discharges, especially of the electrostatic injection device, made it impossible to tune the accelerator to sufficiently high beam currents. The injection device had to be replaced several times due to damage to the insulators supporting the cathode, caused the discharges. RF-power decoupled from the new copper cavity was causing the problems. Two different effects were determined to induce the discharges. On the one hand, RF-power decoupled from the cavity is absorbed by the electrodes of the electrostatic element which leads to the accumulation of charge on the electrodes, creation of halos, and secondary electron emission. In 2014 on the other hand, the high amount of short interruptions was mainly caused by plasma phenomena in the Ring cyclotron. The decoupled RF-power from the flattop cavity resonantly excites secondary electron emission in between the magnet poles of the neighbouring sector magnet. These electrons in turn hit the surface of the trim coils of the magnet and produce ions that stray in the vacuum chamber and are attracted by the electric field of the electrostatic elements. This leads to vapor deposition of conductive material on the insulators that support

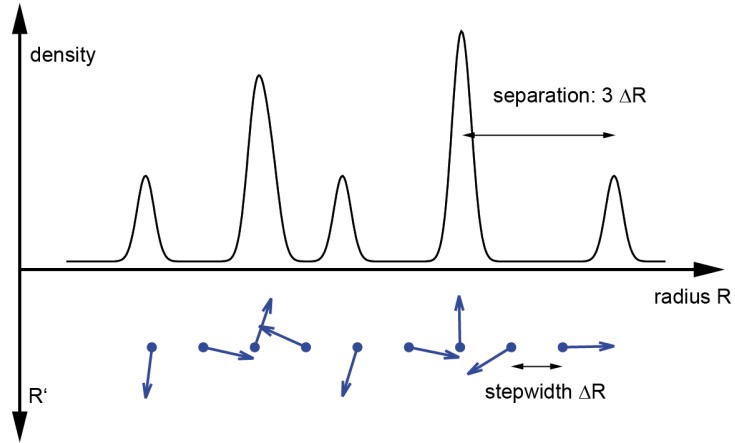

Figure 2.6: Principle scheme of maximizing the beam separation at extraction by utilizing betatron oscillations of the beam center. Important is only the relation between turn-separation and beam width. The 'stepwidth' $\Delta R$ is the distance between turns for betatron amplitude zero. The upper plot shows the beam density along the radius, which is a superposition of Gaussian profiles. In the lower half, the clockwise-rotating phase space vector of the centroid of the beam is shown for each turn. The reduction of the radial tune to $\approx 1.5$ on the last turns is essential for the intended operation of this scheme.

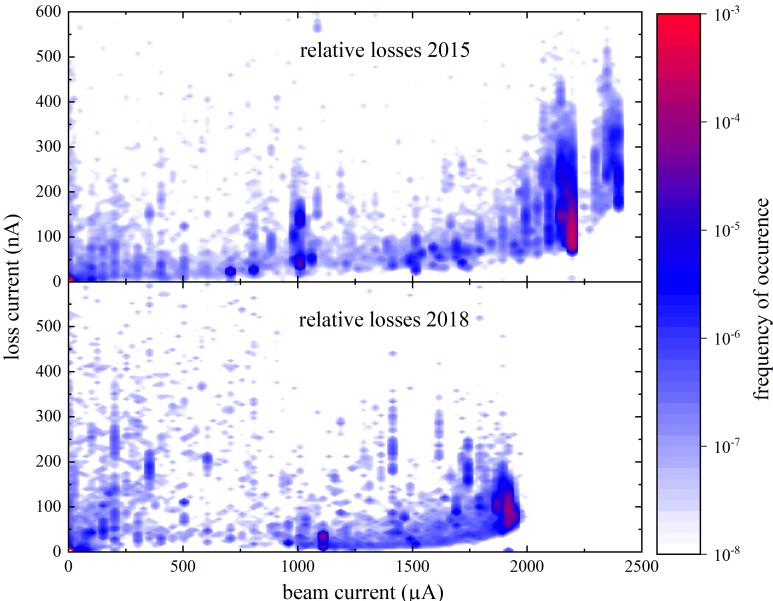

Figure 2.7: Relative losses in the Ring cyclotron during two different operation scenarios. The upper graph depicts the relative losses during the operation in 2015 with a beam current of 2.2 mA for standard operation and 2.4 mA for beam development shifts, respectively. The average loss current at 2.4 mA is approx. 230(44) nA and thus two times higher than at 2.2 mA. Due to the Injector II upgrade, the beam current was limited to 2.0 mA in 2018. The average losses at this current are approx. 82(25) nA

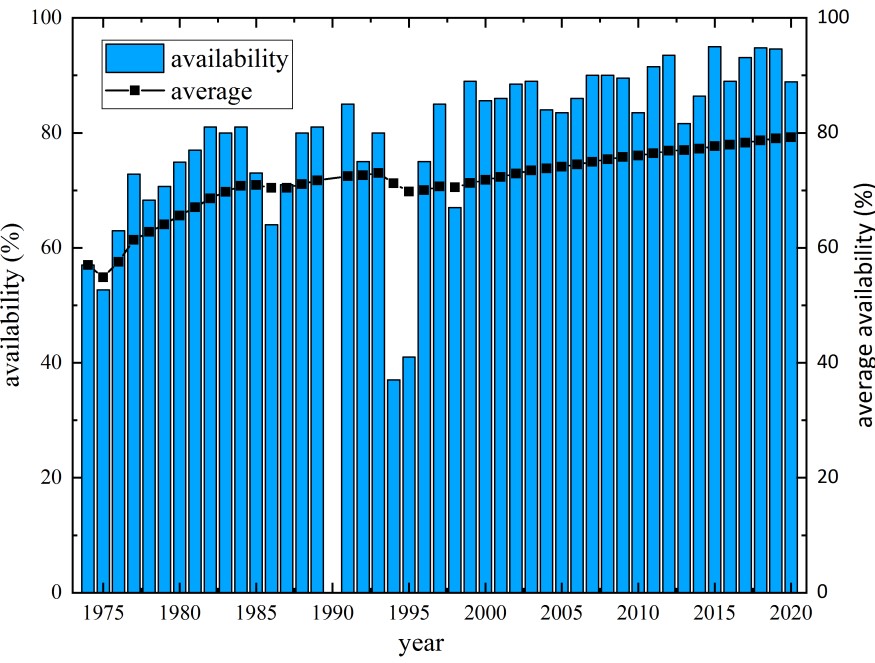

Figure 2.8: Availability of the high intensity proton accelerator facility for the years from 1974 to 2020. The black curve represents the average availability.

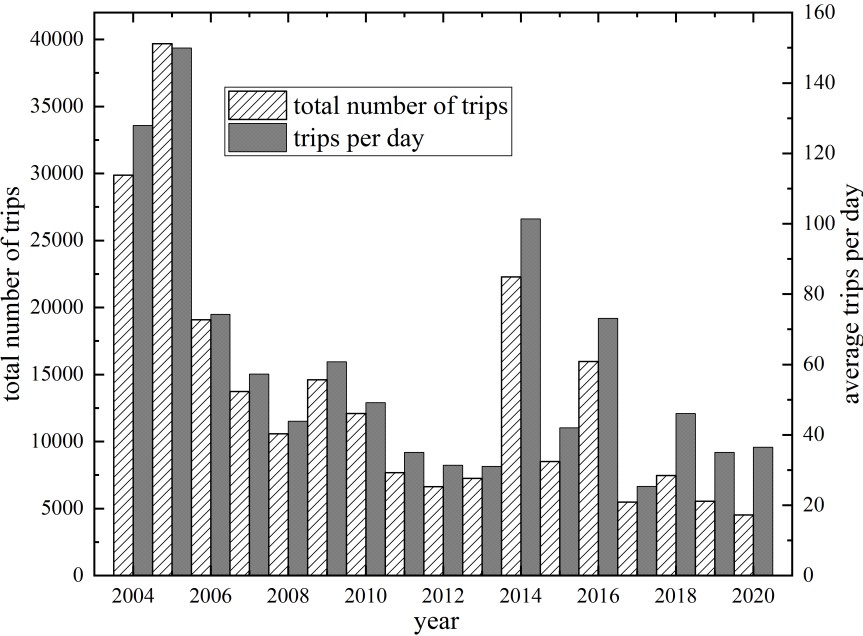

Figure 2.9: Total number of short interruptions for the years 2004 to 2020 (hatched). The solid bars denote the relative number of short interruptions normalized to the number of scheduled days of user operation, i.e., average number of short interruptions (< 5 min.) per day.

the cathode and thus discharges of the electrostatic elements. To mitigate this effect, an aluminium shroud was attached to the electrostatic devices to shield the RF-power and screen it from straying ions.

Though recovery from a discharge of the electrostatic elements may occur in much shorter time, the automatic ramping up of the accelerators lasts between 20 to 30 seconds. Therefore, short interruptions may have a non-negligible impact on the yearly availability. Assuming an average of 15000 short interruptions per year the aggregate downtime constitutes approximately 80 hours. Given 5000 hours of user operation, this results in a loss of availability of 1.6 %

In Figure 2.10, the accumulated outage characteristics for 2004 through 2020 are shown. The most prominent events causing outages are site cooling (15 %), radio frequency systems (13 %), and targets (12 %). Although this does not reflect the characteristics related to each year of operation, it is a guideline for risk management and stock-keeping of spares.

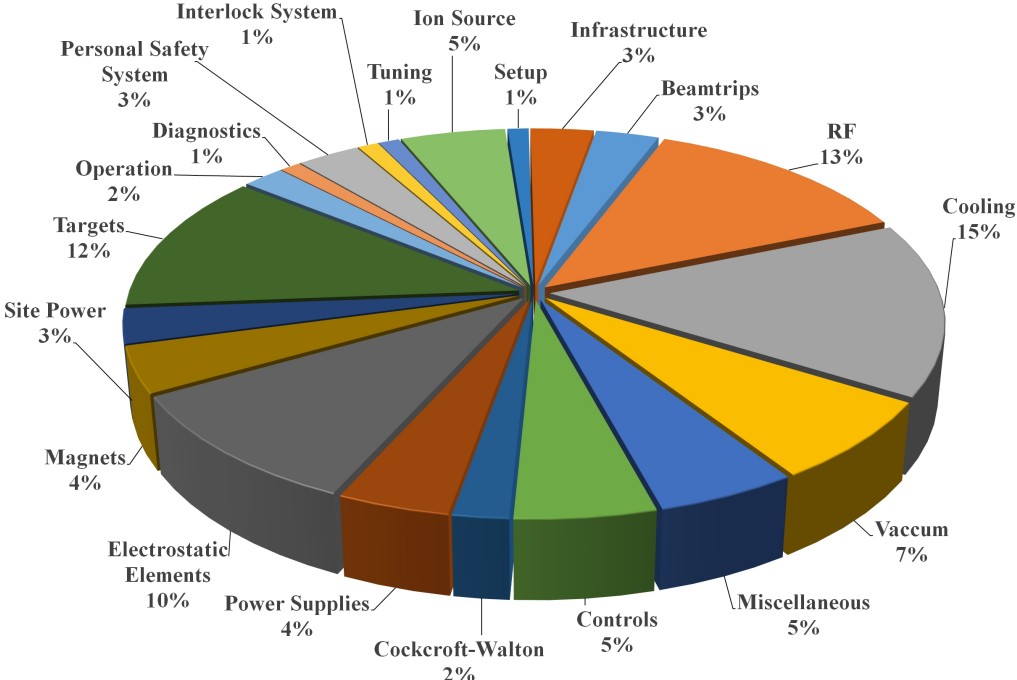

Figure 2.10: Accumulated outage characteristics for the High Intensity Proton Accelerator facility for the years 2004 to 2020.

## 2.6 Grid Power Consumption and Energy Efficiency

The experiments at HIPA require highest intensity particle beams for precise measurements. Producing a megawatt proton beam requires the consumption of several megawatts of electrical power. The goal of further upgrades will be to achieve higher particle flux, rates, and brightness, which will require even greater power. Concurrently, the growing global energy consumption challenges the energy efficiency of any technology including accelerator-driven research facilities. Inevitably, a discussion on improving the energy efficiency of the existing facility presents itself. In this section, the energy efficiency of HIPA will be discussed in detail. Furthermore, it will be shown that by increasing the beam power an even higher energy efficiency may be achieved.

Figure 2.11 shows the power consumption break down of the proton facility. The overall power consumption of the facility in routine operation at 2.2 mA beam current is approxi-

mately 12.5 MW. The 5.4 MW of the RF-to-beam power conversion dominates the power consumption. This value scales roughly linearly with beam power (see Figure 2.12): the power consumption of the magnets and auxiliary systems, e.g., cooling, conventional systems, and instruments is virtually independent of the beam power.

With a beam power of up to 1.3 MW and a total power consumption of 12.5 MW, the energy efficiency of the facility is 11 %. This does not reflect the energy efficiency of the bare accelerator, as all experimental facilities (IP2, UCN, SINQ, and all secondary beamline experiments) that require electrical power contribute to the total power consumption. In a detailed study [45], the power consumption of each subsystem (RF-System, magnets, and infrastructure) required only for beam production, was analyzed. According to this study, a minimum of 7.12 MW of power from the power grid is required for a beam current of 2.2 mA. Thus, the energy efficiency of the bare accelerator is 18 %. One might expect the energy efficiency of the facility to increase linearly with beam power, corresponding to the linear behavior of the RF- to beam power conversion denoted in Figure 2.12. However, the power consumption $P_{\mathrm{RF}}$ of the RF-System was measured as a function of the beam current keeping the voltage of the accelerating cavities constant (850 kV per cavity). According to the empirical law of Joho [36] the number of turns in a cyclotron has to be reduced to achieve higher beam currents for constant beam losses. This, in turn, is only possible by increasing the peak voltage $V_{\mathrm{acc}}$ of the accelerating cavities. Since the wall losses $P_{\mathrm{loss}}$ in a cavity scale with $V_{\mathrm{acc}}^2/2R$ (where $R$ is the shunt impedance of the cavities), correspondingly more electrical power is needed to increase the beam current. Since $P_{\mathrm{RF}} = P_{\mathrm{loss}} + k \cdot P_{\mathrm{beam}}$ where $k$ characterizes the efficiency of the RF-amplifier chain, this results in a non-linear behavior of the RF- to beam power conversion. The considerations in the following section will proof that increasing the beam current by reducing the number of turns in the cyclotron will nevertheless increase the energy efficiency of the accelerator facility.

The efficiency $\eta_{\mathrm{acc}}$ of the bare accelerator is defined as the ratio of the beam power $P_{\mathrm{beam}}$ and the total power $P_{\mathrm{tot}}$ needed to operate the accelerator. In a simplified model, $P_{\mathrm{tot}}$ is $P_{\mathrm{loss}} + k \cdot P_{\mathrm{beam}} + P_{\mathrm{aux}}$. The power consumption $P_{\mathrm{aux}}$ of the magnets and auxiliary system, e.g., cooling, conventional systems, and instruments is virtually independent of the beam power. Therefore, the efficiency of the accelerator is

$$\eta_{\mathrm{acc}} = \frac{P_{\mathrm{beam}}}{P_{\mathrm{loss}} + P_{\mathrm{aux}} + k \cdot P_{\mathrm{beam}}} \, . \tag{2.3}$$

As the maximum current $I_{\mathrm{max}}$ extracted from a cyclotron is proportional to $1/N^3$ [36], the number of turns $N$ is

$$N = \frac{E_{\mathrm{kin}}}{q \cdot V_{\mathrm{acc}} \cdot N_c} \, , \tag{2.4}$$

where $N_c$ is the number of cavities and $E_{\mathrm{kin}}$ is the gain in energy of the particles and $q$ their charge. Thus

$$I_{\mathrm{max}} \propto \frac{q^3 \cdot V_{\mathrm{acc}}^3 \cdot N_c^3}{E_{\mathrm{kin}}^3} \text{ and } V_{\mathrm{acc}} = \epsilon \cdot \frac{E_{\mathrm{kin}}}{q \, N_c} \cdot I_{\mathrm{max}}^{1/3} \, , \tag{2.5}$$

where $\epsilon$ is a constant factor. The efficiency of the accelerator as a function of the beam current can then be deduced to be

$$\eta_{\mathrm{acc}} \approx \frac{E_{\mathrm{kin}} \cdot I}{\frac{\epsilon^2 \cdot E_{\mathrm{kin}}^2}{2 \cdot N_c \cdot Z \cdot q} \cdot I_{\mathrm{max}}^{\frac{2}{3}} + k \cdot E_{\mathrm{kin}} \cdot I + q \cdot P_{\mathrm{aux}}} \, . \tag{2.6}$$

As the denominator contains the beam current with an exponent of $\leq 1$ the efficiency will increase with the beam current. With the actual setup of the Ring cyclotron, i.e., cavity voltages

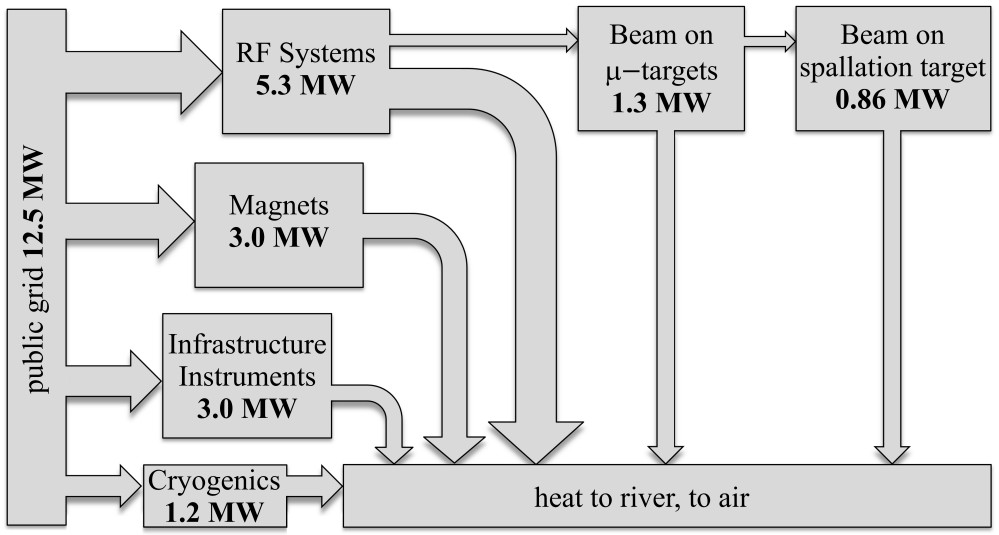

Figure 2.11: Breakdown of the power flow in the Proton Accelerator facility for a beam current of 2.2 mA.

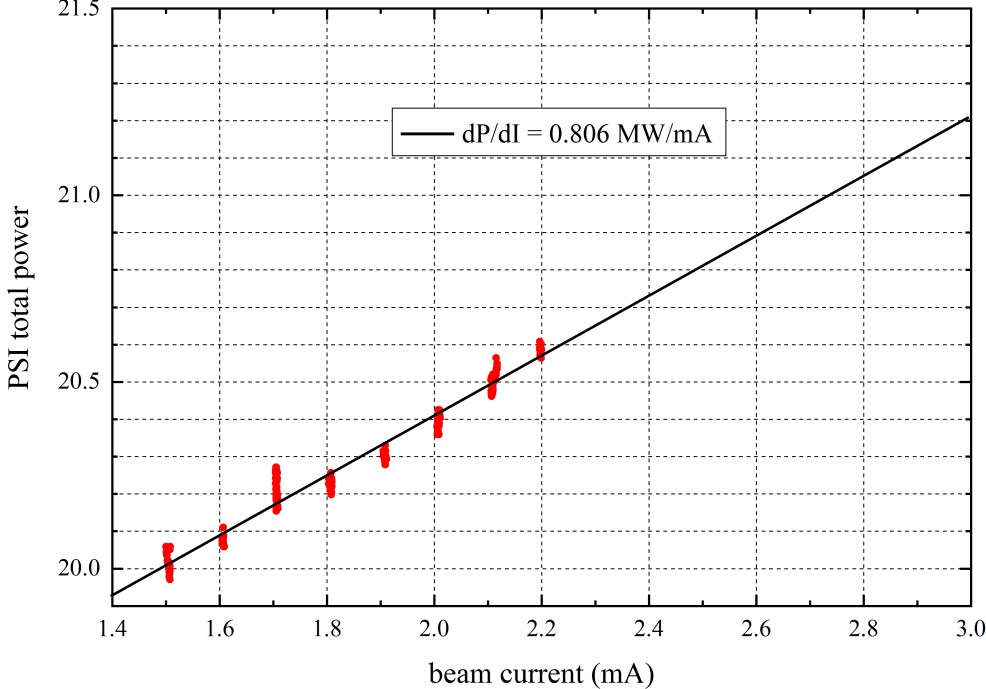

Figure 2.12: Grid to beam power conversion as a function of the beam current. The measurements (red) where recorded with a fixed cavity voltage of each 850 kV. The black line denotes a linear regression of the data. Extrapolated to 3 mA of beam current, a power of 21.2 MW from the grid would be needed.

of $V_{\text{acc}} = 850\,\text{kV}$ and a beam current of $2.4\,\text{mA}$, the efficiency is 0.18, which is the highest for any high power accelerator existing to date [46]. By increasing the beam current to the ultimate goal of $3.0\,\text{mA}$ at a cavity voltage of $1\,\text{MV}$ an efficiency 0.21 could be achieved. This is feasible at PSI, since the RF-system is designed for a peak voltage of up to $1.2\,\text{MV}$. The limitation of $850\,\text{kV}$ and thus the maximum beam current is given by the flattop cavity system. Currently, the maximum flattop voltage is $550\,\text{kV}$ corresponding to the necessary $11\,\%$ of the main cavity voltage. For an operation at higher voltages the flattop system, including the cavity and the amplifiers, would have to be replaced. It is important to note, that these values are valid for the specific setup of the Ring cyclotron, i.e., four accelerating cavities with a given shunt impedance $R$. If the acceleration voltage or the energy gain per turn respectively were distributed among 8 cavities, the wall losses per cavity would be lower. If calculated for eight cavities, the efficiency would be 0.2 at $2.4\,\text{mA}$. It is obvious that the shunt impedance $R$ is one of the main parameters to optimize the efficiency at a given gap voltage. In fact, the shunt impedance only depends on the geometry and choice of material of the cavity and is, therefore, the parameter to optimize. This is an important consideration for future cyclotron based accelerator driven systems.

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
