# Peer review of "The High Intensity Proton Accelerator Facility"

_SciPost Physics Proceedings, doi:SciPost Phys. Proc. 5, 002 (2021)_

## Round 1 · Referee Report · Adrian Signer (Referee 1) · 2021-7-20

Report

We (the editors Cy Hoffman, Klaus Kirch, Adrian Signer) had the opportunity to review an earlier draft of the
article and were in communication with the authors before the submission. All our comments and
suggestions have been taken into account. Hence, we think the paper can now be published in the
current form.

---

## Round 1 · Referee Report · Anonymous (Referee 2) · 2021-7-27

Report

This paper is a gem and should be published after the respective corrections/imporivements

Requested changes

1) Line 10ff: instead of neutron in a spallation target ... --> neutrons in spallation targets and pions in meson production targets. The pions decay into muons and neutrinos. Pions and muons are used in condensed ...

2) Line 19: According to the Style Manual of the American Physical Society one should write: ... tens of microampere.

3) Line 27: To my knowledge, the maximal extracted current of Injector I was 180 $\mu$A, see also Fig. 24 before 1985, where the power is 89 kW corresponding to about 151 $\mu$A.

4) Footnote page 1: Throughout the paper the name ,,Ring cyclotron" was used while in the footnote it says ,,ring Cyclotron"

5) Line 34: The statement ,,Injector I was not able to deliver beam currents higher than originally specified" is wrong! The specified maximal beam current was 100 $\mu$A, while the maximal delivered beam current for longer periods (1984 and 1985) was 180 $\mu$A.

6) Line 52: Leave away ("Dees"). The resonators in Injector two do not look at all like Dees.

7) Line 55: Instead of ... particles' almost circular paths... --> ...particles on almost circular paths ... In fact the particle move on almost spiral paths!

8) Line 58: Give reference to a paper ,,The Meson Production Targets in the high energy beamline of HIPA at PSI", D. Kisselev, P.A. Duperrex, S. Jollet, D. Laube, D. Reggiani, R. Sobbia, V. Talanov, These Proceedings (2021).

9) Line 59: ... the beam loses only a small fraction of its energy.

10) Line 65: The references [13-16] are not the main publications to describe the PSI UCN source. It is mandatory to add: The PSI ultra-cold neutron source, A. Anghel, F. Atchison, B. Blau, B. van den Brandt, M. Daum, R. Doelling, M. Dubs, P.-A. Duperrex, A. Fuchs, D. George, L. Goeltl, P. Hautle, G. Heidenreich, R. Henneck, S. Heule, T. Hofmann, S. Joray, M. Kasprzak, K. Kirch, A. Knecht, J.A. Konter, T. Korhonen, M. Kuzniak, B. Lauss, A. Mezger, A. Mtchedlishvili, G. Petzold, A. Pichlmaier, D. Reggiani, R. Reiser, U. Rohrer, M. Seidel, H. Spitzer, K. Thomsen, W. Wagner, M. Wohlmuther, G. Zsigmond, J. Zuellig, K. Bodek, S. Kistryn, J. Zejma, P. Geltenbort, C. Plonka, S. Grigoriev, Nucl. Instrum. and Meth. in Phys. Res. A 611, 272 (2009)

and

Neutron Optics of the PSI ultracold-neutron source: characterization and simulation, G. Bison, B. Blau, M. Daum, L. Goeltl, R. Henneck, K. Kirch, B. Lauss, D. Ries, P. Schmidt-Wellenburg G. Zsigmond, Europ. Phys. J. A 56, 33 (2020).

11) Line 65: Give reference to a paper: A fast kicker magnet for the PSI 600 MeV proton beam to the PSI ultra-cold neutron source, D. Anicic, M. Daum, G. Dziglewski, D. George, M. Horvath, G. Janser, F. Jenni, I. Jirousek, K. Kirch, T. Korhonen, R. K\"unzi, A. C. Mezger, U. Rohrer, L. Tanner, Nucl. Instrum. and Meth. in Phys. Res. A 541, 598 (2005).

12) Line 122: Please explain the difference between 1 mA from the Injector II and 310 $\mu$A in the Ring cyclotron! as ist is written it suggests that 69 % of the beam gets lost!

13) Line 142: The shut downs start actually in December, right before Christmas!

14) The maximal available pion momentum is about 450 MeV/c. Above that momentum, the pion production cross section tends to zero.

15) Line 190: The facility, originally designed for ...

16) Line 205: My suggestion: Following the installation of the last (fourth) new copper cavity...

17) Line 218: ... few mSv/h --> ... a few millisievers per hour...

18) Line 219: ... increase the beam current --> ...increase of the beam current.

19) Line 227: ... the radial tune and U_t the enery gain ... --> ... the radial tune, U_t the energy gain...

20) Line 263 and throughout of the paper: aluminum is American English, European English is aluminium

21) Figure caption 2.8: 1974 and 2020 --> 1974 to 2020

22) Line 304: The 5.4 MW of the RF-to-beam... add "of"!

23) Line 305: ... roughly linear --> roughly linearly

24) Line 357: ... of eth zurich --> of ETH Zurich

---

## Round 2 · Author Response

We are thank the referees for their valuable comments and corrections.

---

## Round 2 · List of Changes

The reviewer suggested the following changes:

1) Line 10ff: instead of neutron in a spallation target ... --> neutrons in spallation targets and pions in a meson production target. The pions decay into muons and neutrinos. Pions and muons are used in condensed ...

Done.

2) Line 19: According to the Style Manual of the American Physical Society one should write: ... tens of microampere.

Done.

3) Line 27: To my knowledge, the maximal extracted current of Injector I was 180 $\mu$A, see also Fig. 24 before 1985, where the power is 89 kW corresponding to about 151 $\mu$A.

Thanks for pointing this out. Indeed 180 uA had been reached.

4) Footnote page 1: Throughout the paper the name ,,Ring cyclotron" was used while in the footnote it says ,,ring Cyclotron"

Corrected.

5) Line 34: The statement ,,Injector I was not able to deliver beam currents higher than originally specified" is wrong! The specified maximal beam current was 100 $\mu$A, while the maximal delivered beam current for longer periods (1984 and 1985) was 180 $\mu$A.

Corrected.

6) Line 52: Leave away ("Dees"). The resonators in Injector two do not look at all like Dees.

Done.

7) Line 55: Instead of ... particles' almost circular paths... --> ...particles on almost circular paths ... In fact the particle move on almost spiral paths!

Corrected.

8) Line 58: Give reference to a paper ,,The Meson Production Targets in the high energy beamline of HIPA at PSI", D. Kisselev, P.A. Duperrex, S. Jollet, D. Laube, D. Reggiani, R. Sobbia, V. Talanov, These Proceedings (2021).

Inserted.

9) Line 59: ... the beam loses only a small fraction of its energy.

Done.

10) Line 65: The references [13-16] are not the main publications to describe the PSI UCN source. It is mandatory to add: The PSI ultra-cold neutron source, A. Anghel, F. Atchison, B. Blau, B. van den Brandt, M. Daum, R. Doelling, M. Dubs, P.-A. Duperrex, A. Fuchs, D. George, L. Goeltl, P. Hautle, G. Heidenreich, R. Henneck, S. Heule, T. Hofmann, S. Joray, M. Kasprzak, K. Kirch, A. Knecht, J.A. Konter, T. Korhonen, M. Kuzniak, B. Lauss, A. Mezger, A. Mtchedlishvili, G. Petzold, A. Pichlmaier, D. Reggiani, R. Reiser, U. Rohrer, M. Seidel, H. Spitzer, K. Thomsen, W. Wagner, M. Wohlmuther, G. Zsigmond, J. Zuellig, K. Bodek, S. Kistryn, J. Zejma, P. Geltenbort, C. Plonka, S. Grigoriev, Nucl. Instrum. and Meth. in Phys. Res. A 611, 272 (2009)

and

Neutron Optics of the PSI ultracold-neutron source: characterization and simulation, G. Bison, B. Blau, M. Daum, L. Goeltl, R. Henneck, K. Kirch, B. Lauss, D. Ries, P. Schmidt-Wellenburg G. Zsigmond, Europ. Phys. J. A 56, 33 (2020).

Thank you for the pointing to these references.

11) Line 65: Give reference to a paper: A fast kicker magnet for the PSI 600 MeV proton beam to the PSI ultra-cold neutron source, D. Anicic, M. Daum, G. Dziglewski, D. George, M. Horvath, G. Janser, F. Jenni, I. Jirousek, K. Kirch, T. Korhonen, R. K\"unzi, A. C. Mezger, U. Rohrer, L. Tanner, Nucl. Instrum. and Meth. in Phys. Res. A 541, 598 (2005).

Done.

12) Line 122: Please explain the difference between 1 mA from the Injector II and 310 $\mu$A in the Ring cyclotron! as ist is written it suggests that 69 % of the beam gets lost!

Changed to: "Ten years later, after the first commissioning of the new pre-accelerators, a beam current of $1\,\mathrm{mA}$ was achieved with Injector~II alone, and $310\,\rm{\mu A}$ in combination with the Ring cyclotron."

13) Line 142: The shut downs start actually in December, right before Christmas!

Changed.

14) The maximal available pion momentum is about 450 MeV/c. Above that momentum, the pion production cross section tends to zero.

Changed.

15) Line 190: The facility, originally designed for ...

Changed.

16) Line 205: My suggestion: Following the installation of the last (fourth) new copper cavity...

Changed to: "Following the installation of the fourth and last new copper cavity..."

17) Line 218: ... few mSv/h --> ... a few millisievers per hour...

Changed.

18) Line 219: ... increase the beam current --> ...increase of the beam current.

Changed.

19) Line 227: ... the radial tune and U_t the enery gain ... --> ... the radial tune, U_t the energy gain...

Changed.

20) Line 263 and throughout of the paper: aluminum is American English, European English is aluminium

Changed to British.

21) Figure caption 2.8: 1974 and 2020 --> 1974 to 2020

Changed.

22) Line 304: The 5.4 MW of the RF-to-beam... add "of"!

Changed.

23) Line 305: ... roughly linear --> roughly linearly

Changed.

24) Line 357: ... of eth zurich --> of ETH Zurich

Changed.

---

## Editorial Decision

published